# A Marine λ-Oligocarrageenan Inhibits Migratory and Invasive Ability of MDA-MB-231 Human Breast Cancer Cells through Actions on Heparanase Metabolism and MMP-14/MMP-2 Axis

**DOI:** 10.3390/md19100546

**Published:** 2021-09-28

**Authors:** Rémi Cousin, Hugo Groult, Chanez Manseur, Romain Ferru-Clément, Mario Gani, Rachel Havret, Claire Toucheteau, Grégoire Prunier, Béatrice Colin, Franck Morel, Jean-Marie Piot, Isabelle Lanneluc, Kévin Baranger, Thierry Maugard, Ingrid Fruitier-Arnaudin

**Affiliations:** 1BCBS Group (Biotechnologies et Chimie des Bioressources pour la Santé), Laboratoire Littoral Environnement et Sociétés, La Rochelle University, UMR CNRS 7266, 17000 La Rochelle, France; remi.cousin1@univ-lr.fr (R.C.); hugo.groult@univ-lr.fr (H.G.); chanez.manseur@univ-lr.fr (C.M.); romain.ferruclement@univ-lr.fr (R.F.-C.); mario.gani@etudiant.univ-lr.fr (M.G.); rachel.havret@etudiant.univ-lr.fr (R.H.); claire.toucheteau@hotmail.fr (C.T.); grg.prunier@orange.fr (G.P.); beatrice.colin@univ-lr.fr (B.C.); jean-marie.piot@univ-lr.fr (J.-M.P.); isabelle.lanneluc@univ-lr.fr (I.L.); thierry.maugard@univ-lr.fr (T.M.); 2Laboratoire Inflammation, Tissus Epithéliaux et Cytokines, Poitiers University, LITEC EA 4331, 86073 Poitiers, France; f.morel@univ-poitiers.fr; 3Aix-Marseille University, CNRS, INP, Inst Neurophysiopathol, 13385 Marseille, France; kevin.baranger@univ-amu.fr

**Keywords:** polysaccharide, oligosaccharide, heparin, λ-carrageenan, heparanase, metalloproteinase, breast cancer, MMP-14, MDA-MB-231, shRNA

## Abstract

Sugar-based molecules such as heparins or natural heparan sulfate polysaccharides have been developed and widely studied for controlling heparanase (HPSE) enzymatic activity, a key player in extracellular matrix remodelling during cancer pathogenesis. However, non-enzymatic functions of HPSE have also been described in tumour mechanisms. Given their versatile properties, we hypothesized that sugar-based inhibitors may interfere with enzymatic but also non-enzymatic HPSE activities. In this work, we assessed the effects of an original marine λ-carrageenan derived oligosaccharide (λ-CO) we previously described, along with those of its native counterpart and heparins, on cell viability, proliferation, migration, and invasion of MDA-MB-231 breast cancer cells but also of sh-MDA-MB-231 cells, in which the expression of HPSE was selectively downregulated. We observed no cytotoxic and no anti-proliferative effects of our compounds but surprisingly λ-CO was the most efficient to reduce cell migration and invasion compared with heparins, and in a HPSE-dependent manner. We provided evidence that λ-CO tightly controlled a HPSE/MMP-14/MMP-2 axis, leading to reduced MMP-2 activity. Altogether, this study highlights λ-CO as a potent HPSE “modulator” capable of reducing not only the enzymatic activity of HPSE but also the functions controlled by the HPSE levels.

## 1. Introduction

Triple negative breast cancer (TNBC), defined as breast cancer cells lacking expression of oestrogen receptor, progesterone receptor, and human epidermal growth factor receptor 2 (HER2), accounts for up to 20% of all breast cancer. TNBC status is associated with a worse prognosis and a more aggressive natural history than breast cancers that are hormone receptor and/or HER2 positive [1]. Patients with TNBC do not benefit from hormonal or trastuzumab-based therapies. Thus, surgery, radiotherapy, and chemotherapy, either individually or in combination, remain the most popular treatments.

In this context, the most widely used drug treatments against TNBC are still anti-mitotic-based chemotherapy directed against cell proliferation, despite strong drawbacks such as limited durability of tumour response to the treatment and associated toxicities [1]. This approach underestimates the gain of the invasive/metastatic phenotype, which is a key feature for such cancers [2] and accounts for more than 90% of mortality [3]. Given the lack of targeted treatments against metastatic TNBC in clinic [4], the effect of immune checkpoint inhibitors [5] or specific agents like PARP and AKT inhibitors have been evaluated. In addition, antimetastatic drugs, including migrastatic compounds interfering with all modes of invasion of cancer cells, are considered a new option to stop the natural course of the disease [4]. Several antimetastatic molecules have already been shown to be effective but toxicity concerns have impaired their use in clinic [4]. In the context of invasive cancer such as TNBC, it is therefore necessary to develop new molecules with better innocuity, while maintaining migrastatic effects. Thus, biomolecules targeting the specific mechanisms responsible for TNBC aggressiveness, namely migration and invasion abilities, are of great interest [6].

Cancer research has stressed the importance of the tumour microenvironment (TEM) in the progression of the disease, particularly in ensuring the mechanisms of migration and invasion [7]. Indeed, the TEM undergoes biochemical changes leading to new cancer-enhancer properties. For example, the over-expression and -secretion of hydrolytic enzymes such as glycosidases and proteinases, either from cancerous or neighbouring cells, modulate the extracellular matrix (ECM) composition and structure during tumour progression [8]. Among these enzymes, heparanase (HPSE thereafter), the only known β-D-glucuronidase capable of degrading specific glycosidic bonds in heparan sulfate (HS) side chains present on heparan sulfate proteoglycan (HSPG) [9], modulates HSPG levels and their functions [10]. Through its hydrolytic action, HPSE facilitates direct tumour cells migration and invasion [10]. Clinical studies have shown that higher HPSE levels correlate with higher tumour growth rates, enhanced angiogenesis, development of metastases, resulting in the worst prognosis for patients [11,12]. Targeting HPSE has already been shown to efficiently reduce cancer progression in patients [11,13]. In addition to this deleterious enzymatic activity, HPSE also shows non-enzymatic roles, notably in gene regulation, exosome formation and autophagy that are known to be implicated in many metabolic pathways of cancerous cells [14]. HPSE biology, especially, appears to be intimately linked to matrix metalloproteinases (MMPs) [15], other key actors in cancerous cell migration and invasion processes [16]. For example, in myeloma, HPSE drives aggressive tumour phenotype by up-regulating MMP-9 expression and activity within its TEM [17]. HPSE has also been reported to control expression of MMPs such as MMP-2 and -14, but without clear elucidation of this interplay [15].

Various drugs targeting HPSE activity have already been developed and some of them are currently assessed in clinical trials [18] to reduce cancer progression, either as adjuvant or as novel therapy [19]. One important family of these drugs are structural mimetics of HS derived from heparin, an endogenous sulphated polysaccharide (PS) produced by mast cells and proposed as an inhibitor of HPSE enzymatic activity [20]. However, due to its potent and well-known anticoagulant properties, heparin can induce side-effects, including anticoagulant activity leading to internal bleeding. Its use as a cancer therapy is therefore challenging for patients [21]. Furthermore, clinically used heparins are mainly derived from porcine intestine mucosa with very low yield of extraction and are chemically difficult to synthesize. As an alternative to heparin and its derivatives, we have previously described that natural PS from non-animal origin such as fucoidans, carrageenans, and dextran sulphate that are structurally related to HS/heparin, have anticancer potential [22]. We notably generated and selected a 5.9 kDa low molecular weight oligomer of λ-carrageenan (λ-CAR), namely λ-CO, presenting anti-HPSE in vitro activity and potential anti-tumour activities by reducing angiogenesis and cell migration of breast cancer cells [23,24]. However, the exact contribution of HPSE in these beneficial effects of λ-CO on these cells remained to be elucidated.

Biological studies that have already investigated oligosaccharides (OS)-based HPSE inhibitors showed different results and complex mechanisms depending on the cellular or in vivo models used, not only with regard to the inhibition of HPSE enzymatic activity [18]. For example, Ritchie et al. showed that Roneparstat, a well-known HPSE inhibitor, was able to inhibit HPSE activity and downregulate proangiogenic factor expression, such as VEGF, but also regulate MMP-9 expression in myeloma model in vivo [25]. HPSE can also regulate autophagic mechanism of cancer cells. Indeed, HPSE inhibition by PG545 has been shown to downregulate autophagy by regulating LC3 expression, a well-known marker of autophagy, in tumour xenograft models of human pharyngeal carcinoma [26]. The multiple molecular mechanisms associated with this class of biomolecules may be explained by the specific features of these multitasking sugar compounds, advantageous because they can have multiple pharmacological activities and many possible interactions, but difficult to decipher at the molecular/cellular level.

In this context, we wanted to deepen our understanding of the biological mechanisms of the λ-CO underlying its potential anti-tumour activity, focusing on the HPSE biology in the MDA-MB-231 TNBC cell line. In order to appreciate if the effects depend on the molecular weight (MW), the length, or the saccharide composition, we included in the study the native λ-CAR counterpart of λ-CO as well as two heparin standards, namely the native unfractionated form (UFH) and one low molecular weight heparin derivative (LMWH) [27]. Given they display different IC50 to inhibit in vitro HS degradation by HPSE, we were also able to assess the real input of HPSE-enzymatic inhibition related to the cell effects of these sugar-based inhibitors [23,24]. We assessed the effects of these four sugars on the viability, proliferation, migration, and invasion properties of the MDA-MB-231 and sh-MDA-MB-231 cells, in which the expression of HPSE was downregulated with shRNA treatment. Moreover, to widen the explanation of the effects to the sole HPSE activity inhibition, we investigated other basic mechanisms involved by evaluating levels of HPSE and MMPs using RT-qPCR, zymography and Western blot (WB) analyses. We showed here that λ-CO was the best candidate to reduce cell migration and invasion, in a HPSE-dependent manner. Moreover, we evidenced that these inhibitory effects were associated not only with inhibition of HPSE activity but also with a tight control of MMP-2 levels and activity through downregulation of MMP-14 (MT1-MMP) expression. Finally, we validated these results in non-invasive MCF-7 breast cancer cells line that harbours a distinct HPSE expression profile, i.e., with lower HPSE implication [28].

## 2. Results and Discussion

### 2.1. Evaluation of Anti-HPSE Heparin (UFH), λ-Carrageenan (λ-CAR) and Their Low Molecular Weight Derivatives LMWH and λ-CO on MDA-MB-231 Cells Viability

#### 2.1.1. Physicochemical Properties and Heparanase IC50 Values

In previous studies [22,27], we described the production of LMWH of 5.1 kDa by the radical depolymerisation technic from an unfractionated commercial heparin (UFH), a standard for in vitro enzymatic inhibition of HPSE [22,27]. Using the same depolymerisation strategy, we produced a low molecular weight carrageenan oligomer (λ-CO) as a new marine-derived HPSE inhibitor from λ-CAR extracted and purified from red algae [24]. The physicochemical properties of this original λ-CO compared to its native PS counterpart and the heparin standards are displayed in Table 1. UFH and LMWH were stronger HPSE inhibitors than λ-CO based on their respective IC50 values, while λ-CO showed the best anti-tumour activities in a model of MDA-MB-231 cancer cell migration [24].

#### 2.1.2. Anti-HPSE Sugars Display No Toxicity and No Proliferative Effect on MDA-MB-231 Cells

In order to determine the potential effects of each sugar-based molecule on cell viability, a MDA-MB-231 cell line was examined using MTT assay. None of the concentrations tested, i.e., 25, 50 (not shown) and 100 µg/mL, induced any change in viability (Figure 1A), whereas 100 µg/mL is a concentration known to efficiently decrease angiogenesis and MDA-MB-231 cell migration [23,24]. In order to clarify the effects of the compounds on cellular growth, we monitored the activities of these molecules on proliferation rate by cell counting, using trypan blue exclusion. After 24 h of treatment, no effects were observed (Figure 1B). This last result is consistent with previous findings showing that λ-CAR is not cytotoxic on HeLa cells even at 2.5 mg/mL after 72 h of treatment [29]. While some authors reported scarce cytotoxicity of λ-CAR at doses above 500 µg/mL [29,30], our results were similar to those reporting the absence of cytotoxicity of native λ-CAR on murine melanoma or murine breast cancer cells at a concentration of 1000 µg/mL [31]. Altogether, these results suggest that the forthcoming biological effects of the sugars on TNBC cells will not arise from cytotoxicity.

There are different strategies to fight against tumour progression. The main ones are the use of antimitotic agents acting on cell proliferation to reduce tumour growth; however, they display severe drawbacks [4]. Another strategy we were seeking here consists of blocking invasion mechanisms and metastatic dissemination to reduce tumour aggressiveness, with candidates acting as migrastatic drugs with no toxicity and reduced antiproliferative activities. The results in Figure 1 suggest that none of the sugar molecules studied here have cytotoxic or anti-proliferative effects, two requirements for the successful establishment of new migrastatic agents capable of inhibiting cancer cell motility and invasiveness.

### 2.2. λ-CO Can Reduce Migratory and Invasion Ability of MDA Cells; Comparison with the Non-Invasive MCF-7 Cell Line

#### 2.2.1. Effect of the Anti-HPSE Sugars on Migration and Invasion of MDA-MB-231 Cells

We assessed the effects of the different sugars on the migration and invasion of MDA cells (MDA-MB-231 cells transfected with a scramble shRNA), using a Boyden chamber with FBS as chemoattractant. In order to check if compounds could interact with chemotactic factors in FBS, we first performed a cell-chemotaxis control assay incubating the compounds in the lower chamber with the FBS as chemoattractant (Appendix A). At 100 µg/mL, none of them inhibited the basic chemotactic potential of FBS towards MDA cells and we continued with the incubation of the compounds in the upper chamber. We have previously shown that λ-CO at 100 µg/mL significantly reduced MDA-MB-231 cell migration [24]. Under the same experimental conditions, we showed that UFH inhibited cell migration by ~40% (Figure 2A,B). Despite an apparent drop of 35%, the LMWH effect did not reach significance (*p* = 0.056). Regarding λ-CAR and λ-CO, we found opposite effects. While λ-CAR dramatically increased cell migration by a factor 2, λ-CO decreased it by 60% compared with control conditions. Thus, the depolymerisation form of λ-CAR, i.e., λ-CO, positively reversed the promigratory effects of λ-CAR (Figure 2A,B). OGT 2115, a well-known small chemical HPSE inhibitor with proven anti-migration properties [32] was used as a control to evaluate implication of HPSE enzymatic activity in migratory processes. OGT 2115 significantly inhibited cell migration by 20% confirming implication of HPSE enzymatic activity in such biological mechanisms (Figure 2B). We then assessed their effects on cell invasion. To do so, we added a Matrigel layer on the top of the inserts of the Boyden chamber. Under these conditions, we detected a small, but non-significant invasion inhibiting effect for UFH and none for LMWH (Figure 2C,D). On the other hand, we found profiles like those found in the migration assay for λ-CAR and λ-CO (Figure 2D). Indeed, λ-CAR tended to increase cell invasion by a factor 2, while λ-CO significantly reduced cell invasion by ~37% compared with control conditions. It is well-known that reduction of the size of PS chains modulates biological activities and/or improves their innocuities in vivo [33]. Therefore, optimized depolymerisation methods are often proposed for the production of homogeneous low molecular weight oligosaccharides formulations with more specific/controlled biological properties and in vivo suitability [34]. In the case of heparin, our results showed a neutral impact of depolymerisation on the activities of UFH against MDA migration and invasion. However, we have previously demonstrated that LMWH presents at least reduced anticoagulant activities more suitable for medical application in cancer [22].

For λ-CAR, the results highlighted the high beneficial value brought by depolymerisation with regard to the activity against migration and invasion of MDA cells. On the one hand, λ-CAR showed a strong pro-invasive and migratory influence on this cancer cell line. This might come from the very high MW of λ-CAR (Table 1) and its rheological/viscosity properties that induce cell grouping or serve as scaffold to promote a collective cell migration. Besides, native carrageenan is often described as promising cell-carrier materials or wound dressing matrixes in tissue engineering and wound healing in other applications [35]. This reinforces the low relevance of native λ-CAR as a drug agent targeting the tumour ECM we are investigating in this work. λ-CAR was set aside for the following experiments in this study. On the other hand, λ-CO showed the best results out of the four compounds, underlining its promise as a potential novel agent to fight against migration and invasion of TNBC. The results obtained with OGT 2115, a specific HPSE inhibitor, confirmed that targeting HPSE enzymatic activity is important to reduce MDA cell migration. However, lower effects were observed for the heparin species in comparison to λ-CO, despite a better in vitro anti-HPSE enzymatic activity. This may indicate that at the cellular level, heparins have, at least partially, preferential interactions with other biomolecules unrelated to HPSE and less contributory to the outcome monitored by our cell-based assays. Another possibility would be that λ-CO might act on additional biological pathways and not only on HPSE activity.

#### 2.2.2. Effect of the Anti-HPSE Sugars on Migration and Invasion of Transfected Sh-MDA-MB-231 Cells with HPSE Downregulation

To assess this point, we generated and selected upon puromycin treatment a cell population stably transfected with sh-RNA against HPSE to reduce its expression. In these cells named sh-MDA, expression of *HPSE* gene was reduced by ~70% compared with MDA cells, as measured by RT-qPCR (Figure 3A). The decrease in HPSE levels was further confirmed in whole cell lysates (wcl) by Western blot (WB) with a ~50% reduction in sh-MDA cells compared with MDA cells (Figure 3B).

We studied the effects of HPSE sh-RNA on cell viability and proliferation rate, using MTT and cell counting, respectively. First, we found no effects of sh-RNA treatment on cell proliferation (Figure 3C) but surprisingly, cell viability was reduced by ~50% in sh-MDA cells compared with MDA cells (Figure 3D), indicating that the downregulation of the HPSE levels might affect mitochondrial activity. Indeed, deregulation of the pathway related to mitochondrial activity after modulation of HPSE levels finds support in the literature, in particular for autophagy activity [36]. To validate this assumption, we evaluated levels of LC3, a protein associated with autophagosome formation. As expected, we found lower levels of LC3 in sh-MDA compared to MDA cells (Figure 3E). Autophagy is modulated during tumour progression allowing cancer cells to resist to therapeutic agents at advanced stages [26]. As for STAT3 [37], a downregulation of HPSE levels in MDA cells led to a drop of LC3 levels that renders cells more sensitive to drug delivery using chemotherapy. On the other hand, similarly to MDA, incubation of our molecules with sh-MDA did not modulate the proliferation rate (Figure 3C) or the viability (Figure 3D) after 24 h of treatment, excluding possible toxicity that could have biased the results of the upcoming migration and invasion assays.

As migration and invasiveness are critical steps involved in metastasis and given HPSE plays critical roles in these phenomena [38,39], we investigated these characteristics on sh-MDA cells. Surprisingly, and in contrast to what has already been described in reported cancer cells stably transfected with an HPSE sh-RNA, i.e., gastric cells [40], in our experimental conditions, we did not observe changes in the migrative and invasive capacities of sh-MDA cells, when compared to the parental and the scramble cell line (data not shown). We can speculate that, according to the cancer cell lines, the importance of HPSE in cell metabolism orchestration might be different. When we examined the effects of our four compounds, the two heparin species still decreased cell migration of sh-MDA, as for the scramble cells (Figure 4A,B). Conversely, the inhibitory effect observed with λ-CO treatment on the MDA was lost in sh-MDA cells (Figure 4A,B). As for λ-CO, OGT 2115 treatment was not able to impair cell migration, thereby confirming the implication of HPSE enzymatic activity in this mechanism (Figure 4B). With regard to cell invasion ability, treatment of sh-MDA cells for 24 h with any of the sugars did not produce any significant effect (Figure 4C,D). Thus, once again, anti-invasive effects of the λ-CO candidate on MDA was lost in case of low HPSE level in MDA.

#### 2.2.3. Effect of the Anti-HPSE Sugars on Migration and Invasion of MCF-7 Cells

In order to assess that the effects of λ-CO depend on the amount of HPSE produced by the cells, we analysed protein levels in the MCF-7 breast cancer cell line that are less invasive than the MDA-MB-231 cells [28]. Indeed, we found lower HPSE levels in MCF-7 than in MDA (Figure 5A). Before cell migration experiments, we ensured (as previously) that the sugar-based molecules did not induce any change in MCF-7 cell viability at 100 µg/mL using an MTT assay (Appendix A). Finally, under the same experimental conditions than MDA, we were not able to detect inhibitory activity of λ-CO nor OGT 2115 on MCF-7 migration (Figure 5B). This result underlined an important difference in the effect of λ-CO between these two types of breast cancer cell lines that are characterized by different HPSE levels.

Overall, it is interesting to note that the heparin species displayed a similar or higher effect on sh-MDA migration than on MDA. This suggest that, at the cellular level, they may have different and preferential interactions with distinct cellular partners that are not altered, or even stimulated, consecutively to HPSE downregulation [41]. In fact, it has been demonstrated that heparin is capable of inhibiting breast cancer metastases via an inhibition of the signalling pathway mediated by the cytokines CXCR4/CXCL12 [42]. Heparin is also able to inhibit angiogenesis by inhibiting the expression of VEGF-A in tumour tissue in a mouse model with lung metastases from mammary cancer cells [43]. Importantly, our results demonstrated that λ-CO effects on the TNBC cell line were related to HPSE metabolism. However, it still remains unclear if it originated only from direct inhibition of the HPSE enzymatic activity. Indeed, the IC50 value is much higher for λ-CO than those of UFH or LMWH towards HPSE enzymatic activity (Table 1) but inhibitory effects of λ-CO on MDA cell migration and invasion was higher than for heparins (Figure 2B,E). This suggests that λ-CO may targets not only HPSE enzymatic activity but also other biological non-enzymatic activities of HPSE. To address this question, we analysed the expression of genes involved in migration of MDA cells upon λ-CO treatment.

### 2.3. λ-CO Reduced HPSE Gene Expression and Acted on Its Trafficking in Treated MDA Cells

We first decided to study expression, level, and activation status of HPSE in presence of UFH and λ-CO. By RT-qPCR, we showed a significant reduction in *HPSE* gene expression in cells treated with λ-CO but not with UFH (Figure 6A). Furthermore, none of them induced differences in the expression of the *CTSL* gene, encoding cathepsin L, a lysosomal proteinase responsible for intracellular activation of HPSE [44] (Figure 6B). We therefore evaluated HPSE levels by WB in wcl and supernatants (sup) prepared from MDA cells. Surprisingly, we only found the pro-form in supernatants while the active form of HPSE was only found in wcl. Interestingly, in wcl, UFH and λ-CO shared similar effects on active HPSE, as they both induced a significant decrease of ~70% and ~60%, respectively, compared with the control (Figure 6C,D). No difference was observed in the supernatants between treatments (Figure 6E). We then showed that UFH significantly increased the pro-HPSE sup over HPSE wcl ratio by ~2.3 compared to the control condition, while λ-CO slightly increased it by 1.3 (Figure 6F). These results represent the first evidence that OS-based inhibitors of HPSE may not only target the HPSE enzymatic activity but also its metabolism, such as its own level and eventually its trafficking. However, they were not associated with a specific regulation of cathepsin L expression, known to be a key proteinase in activation of HPSE [44]. We also showed that these regulations in HPSE level and localisation seems to be slightly different depending on the saccharide composition/type of the sugar molecules, even if further experiments are needed. Thus, UFH tended to induce a change in HPSE localisation with a pumping effect of pro-HPSE out of the cells, whereas the effects of λ-CO seemed to be more related to a downregulation of the *HPSE* gene expression. UFH effects were comparable to those already described [45] but, to the best of our knowledge, such results were not described for λ-carrageenan derived oligosaccharides. λ-CO may have a dual role by directly inhibiting HPSE and tuning down its metabolism. Notwithstanding, we cannot minimize the importance of exploring the as-yet unidentified actors or underlying mechanisms involved in the PS/OS effects, especially those of λ-CO, directed against MDA-MB-231 cells motility.

### 2.4. λ-CO Modulates MMP-2 Activity on Treated MDA Cells

Among these potential molecular players, matrix metalloproteinases (MMP) are well known for their implication in cell migration and invasion in MDA-MB-231 cells [46]. In cancer, the increased activities of MMP-2 and -9, or gelatinases A and B, respectively, lead to ECM degradation, favouring the invasion and dissemination of tumour cells to other tissues and metastasis to distant organs. Both gelatinases are also implicated in cancer development and progression through their functions in cell apoptosis, proliferation, and angiogenesis [47,48]. As illustrated by our RT-qPCR data, we did not find any difference between the MDA cells treated with the sugar-based molecules or with the vehicle (control) regarding the *MMP2* and *MMP9* gene expression (Figure 7A,B). As expected, using gelatin zymography, we detected both pro-MMP-2 (72 kDa) and MMP-2 (55 kDa), as well as pro-MMP-9 (96 kDa) activities in the MDA supernatant (Figure 7C). We showed that pro-MMP-9 activity was not affected by UFH treatment and slightly attenuated, without reaching statistical differences, by λ-CO treatment (Figure 7D). However, concerning pro-MMP-2, λ-CO treatment significantly diminished its activity by 42%, unlike UFH (Figure 7E). The most striking differences were observed in active MMP-2. Indeed, while UFH decreased it by 42%, active MMP-2 was barely detectable upon λ-CO treatment, which dramatically reduced it by 86% (Figure 7F).

As both UFH and λ-CO did not modify *MMP2* gene expression while inducing a drop of active MMP-2 levels, we investigated possible post-translational modifications, namely MMP-2 activation. The latter is complex as it involves mainly other MMPs, such as MMP-14 also called MT1-MMP, and regulatory partners such as TIMP-2, a powerful antagonist of MMP-2, -9, and -14 activities [49]. In this context, we first analysed gene expression of *TIMP2* and *MMP14* by RT-qPCR. Expression of *TIMP2* droped by 50% upon λ-CO treatment, but without reaching statistical significance (Figure 8A). Concerning *MMP14* gene expression, it was significantly decreased upon λ-CO treatment but not by UFH (Figure 8B). We thus investigated MMP-14 protein levels by WB and found that they were in accordance with the RT-qPCR results, as λ-CO reduced MMP-14 levels by 25% and not UFH (Figure 8C). Thus, downregulation of MMP-14 by λ-CO seems, among other possible actors at play, to directly affect MMP-2 activation, a well-known interplay [50] and enzymatic activity involved in cancer cell invasiveness. Overall, this suggests that λ-CO, unlike UFH, impacts both HPSE metabolism and MT1-MMP/MMP-2 axis, thus probably contributing to the global effects observed on the migration and invasion properties of MDA cells.

It is known that matrix metalloproteinases (MMPs) are closely related to the HPSE biology [15]. As we previously demonstrated that λ-CO affects HPSE level and localisation, we wanted to study the specific interplay between HPSE levels and these metalloproteinases (MMP-14 and MMP-2) in MDA. For this, we compared *MMP14* gene expression as well as MMP-2 and MMP-9 activities between MDA, sh-MDA (with lower HPSE expression) and MCF-7 (with naturally lower HPSE level). As expected, compared with MDA, sh-MDA expressed significantly less *MMP14* (Figure 9A). Logically, a drastic reduction of active MMP-2 was observed in sh-MDA compared to MDA cells (Figure 9B). This was also the case for the less-invasive MCF-7 cells compared to MDA, with no MMP-2 activity detected (Figure 9D,E). These two results strengthed the hypothesis of a link between HPSE protein levels and MMP-2 activity, i.e., lower HPSE protein levels are correlated with lower MMP-2 activity. Concerning pro-MMP-9 levels, no significant change was observed in sh-MDA and MCF-7, compared to MDA (Figure 9F). HPSE and MMPs relationship is still a matter of debate in the literature. For instance, on the one hand, Zcharia et al. observed in viable HPSE-deficient mice increased levels of MMP-2 in liver, kidney, and mammary glands compared to WT, as well as increased level of MMP-2 activity (although only measured in serum). However, interestingly, and concerning MMP-14, they found in mammary glands the same tendency we reported in our study. Generally, the differences they observed in MMPs expression across organs reinforced the idea that the links between HPSE and MMPs expression and activity should be cautiously interpreted, depending on the cell or tissue studied. In MDA-MB-231, the same cell line we used, they also showed decreased expression of *MMP14*, *MMP2* and *MMP9* genes upon HPSE overexpression contrary to our results. Importantly, this effect seems to be dependent of HPSE enzymatic activity, as inactive HPSE overexpression had no effects [15]. However, the study only looked at the RNA levels and did not confirm this tendency performing HPSE downregulation experiments.

On the other hand, a correlation between decreased levels of HPSE and MMP-2 has also been described in another study. Indeed, Arctigenin, a lignan, inhibited MDA-MB-231 cells migration and invasion by decreasing levels of HPSE and MMP-2 [51]. Here, our results highlight a comparable link between decreased level of HPSE and a MT1-MMP/MMP-2 axis. Overall, further experiments are needed to understand the functional action of HPSE on MMPs and to decipher the role of its enzymatic-dependent vs. independent activities. Unlike MT1-MMP/MMP-2, pro-MMP-9 levels were not affected significatively upon HPSE downregulation. According to the known contribution of MMP-9 to cell mobility, we cannot exclude that MMP-9 could be an important actor for sh-MDA mobility. Last but not least, the λ-CO effects match the interplay we found between HPSE downregulation in sh-MDA and the reduction of MMP-2 activity. Indeed, altogether, our results demonstrated the capacity of λ-CO to control levels of HPSE and consequently of MMP-14 and MMP-2, to inhibit TNBC cell migration and invasion. The neutral effect of UFH may thereby come from the differences observed in the modulation of the HPSE level and trafficking compared to λ-CO.

## 3. Materials and Methods

### 3.1. Reagents

All the chemical reagents used in this study were of analytical grade and purchased from Sigma–Aldrich (Saint-Quentin Fallavier, France). All the products/media required for the cell culture were from Thermo Fischer Scientific (Villebon-sur-Yvette, France), unless otherwise specified. Fetal bovine serum (FBS) was purchased from Dutscher (Brumath, France). λ-carrageenan (λ-CAR) and Unfractionated heparin (UFH, Porcine mucosal heparin sodium salt ≥100 IU/mg, batch 10160803) were provided by FMC Biopolymer (Villefranche-sur-Saône, France) and Alfa Aesar (Thermo Fisher GmbH, Kandel, Germany), respectively. OGT 2115 was purchased from Santa Cruz Biotechnology (Heidelberg, Germany) and resuspended in DMSO. Recombinant active human HPSE was purchased from R&D systems (7570-GH, Bio-Techne, Lille, France).

### 3.2. Physicochemical Characterizations of the Sulfated Polysaccharides, IC50 Determination

Molecular weights of sugars were estimated by size exclusion chromatography (SEC), using two-columns mounted in series (TSK -GEL G5000PW/TSK-GEL G4000PW or TSK-GEL G4000PW/TSK-GEL G3000PW, Tosoh, Japan) with a LC system from Agilent (Santa Clara, CA, USA) coupled to a differential refractometry detector. The number-average molecular weights (Mn), weight-average molecular weights (Mw) and polydispersity index (PI) were estimated using a method previously described [24]. Standard curves were built with Pullulans (Polymer Standards Service GmbH, Mainz, Germany) or Heparin calibrants (Iduron, UK). The inhibition of HPSE enzymatic activity and IC50 determination were assessed using the heparanase assay toolbox (Cisbio Assay, Codolet, France) and human recombinant HPSE (R&D Systems, Minneapolis, MN, USA), as previously described [24].

### 3.3. Cell Culture

The human breast cancer adenocarcinoma cell line MDA-MB-231 (TNBC) and MCF-7 were purchased from the American Type Culture Collection (ATCC, France Office, Molsheim, France) and maintained in Opti-MEM medium supplemented with 2% FBS and 1% penicillin and streptomycin at 37 °C in a 5% °C O_2_ humid atmosphere. The presence of mycoplasma was controlled using the MycoAlert kit (Lonza, Basel, Suisse).

### 3.4. Transfection Protocol

MDA-MB-231 cells were seeded in six-well plates at 2.10^5^ cells/well and incubated overnight at 37 °C under 5% CO_2_. The medium was removed, cells were washed with phosphate buffered saline solution 1X (PBS), and then transfected with 5 μM of plasmids encoding a scramble-shRNA or HPSE-shRNA (sc-40685-SH, Santa Cruz Biotechnology, Clinisciences, Nanterre, France) using the lipofectamine Plus system, according to the manufacturer’s instructions (Thermo Fischer Scientific). After 48 h, the transfected cells were selected using 2.5 µg/mL of puromycin (Santa Cruz Biotechnology) for MDA-MB-231 for 2 weeks. Successful lower *HPSE* gene expression was further confirmed by RT-qPCR and Western blot (WB) analyses. For the sake of clarity, scramble-shRNA selected cell lines are therefore named MDA and selected MDA-MB-231 treated with HPSE-shRNA are named sh-MDA.

### 3.5. RT-qPCR Analyses

In 25 cm^2^ flasks, 1.10^6^ MDA or sh-MDA cells were seeded for 24 h at 37 °C under 5% CO_2_ in Opti-MEM supplemented with 10% FBS. The cell medium was removed, and cells washed with PBS. Cells were detached using a trypsin/EDTA solution and total RNA was extracted with the RNeasy kit (Qiagen, Courtaboeuf, France) according to the manufacturer’s instructions. RNA was quantified using a Lvis plate system (BMG Labtech, Champigny-sur Marne, France). Single-stranded cDNA was synthesized from 1 µg of RNA from each sample using the cDNA qScript kit (Quantabio, VWR International SAS, Fontenay-Sous-Bois, France). Quantitative PCR was performed with a LightCycler 480 System (Roche Applied Science) thermocycler using SYBR Green PCR technology (Quantabio). The primer sequences used in this study are shown in Table 2 and were designed using Primer3 software. Amplifications were carried out in duplicate using 5 µl cDNA, pre-diluted 10 times in RNA-free water, forward and reverse primers (0.2 µM each) and 2× Mastermix (10 µL), in a total volume of 20 µL. The PCR program comprised of 5 min at 95 °C for polymerase activation, followed by 40 cycles at 95 °C for 20 s, 60 °C or 64 °C for 15 s, and 72 °C for 20 s. The melting curves were established by increasing temperature from 60 to 95 °C with a continuous fluorescence measurement. Samples were normalised to independent control housekeeping gene GAPDH and reported according to the ΔΔCT method as RNA fold increase over untreated cells.

### 3.6. Western Blot Analyses

Proteins from whole cell lysates (wcl) were extracted using RIPA lysis buffer (Thermo Fisher scientific) supplemented with a mix of protease and phosphatase inhibitors (Dutscher) and quantified using the Lowry method (Bio-Rad, Marnes-La-Coquette, France). Equal amounts (80 µg) of proteins from the cell lysates or supernatants (sup) were loaded on 10% SDS-page gel (1h30 migration at 100 volt) and transferred onto nitrocellulose membranes using the transblot system (Bio-Rad). Membranes were blocked with 5% half-fat milk solution (*w*/*v*) for 1 h in Tris Buffered Saline containing 0.1% Tween20 (TBS-T) at room temperature and incubated overnight at 4 °C with the following antibodies: anti-HPSE antibody (66226-Ig, 1/400, Proteintech Europe, Manchester, United Kingdom) or LC3 antibody (14600-1-AP, 1/400, Proteintech Europe) and anti-β-actin (13/E5, 1/1000; Cell Signaling, Ozyme, Saint Cyr l’Ecole, France). Membranes were washed three times with TBS-T and then incubated with the appropriate horseradish peroxidase-conjugated secondary IgG antibody (1/1000 to 1/5000, Thermo Fisher Scientific) at room temperature for 1h30. Immunodetection was performed using the Clarity western ECL kit (Bio-Rad) with a Chemidoc Imaging System (Bio-Rad, Marnes-La-Coquette, France). Optical densities were measured using imageJ software (NIH). All optical density plots represent values normalised to loading controls, as indicated in the figure legends.

### 3.7. Gel Zymography

To detect MMP-9 and MMP-2 levels in the supernatants of MDA, sh-MDA, and MCF-7 cells, we used gelatin zymography as previously described [52]. Equal amounts of serum-free supernatants were subjected to 8% SDS-PAGE containing 1 mg/mL gelatin in non-denaturing and non-reducing conditions. After electrophoresis, the gels were washed twice for 30 min in 2.5% Triton X-100 to remove SDS and incubated overnight in activating buffer (50 mM Tris-HCl, with 10 mM CaCl_2_, pH 7.5) at 37 °C. The gels were then stained with 0.1% Coomassie Brilliant Blue R-250 (Bio-Rad) for 30 min and destained with a solution containing 10% acetic acid until clear bands of gelatinolysis appeared on a dark background. Gels were digitized using the Chemidoc Imaging System (Bio-Rad) and optical density measured using the imageJ software (NIH).

### 3.8. MTT Assay

MDA, sh-MDA, and MCF-7 cells were seeded into 96-well plate at 5.10^3^ cells/well in 100 µL Opti-MEM supplemented with P/S 1% and FBS 2% at 37 °C. After 24 h, the plating medium was completed with 100 µL of FBS-free Opti-MEM and the different sugars were added at different concentrations ranging from 25 to 100 µg/mL or not and treated with vehicle (control). After 24 h, 20 µL of 3-(4,5-Dimethylthiazol-2yl)-2,5-diphenyl tetrazolium bromide (MTT) solution (5 mg/mL) was added to each well and the cells were incubated for 4 h at 37 °C. The medium was removed and 200 µL of DMSO were added to each well. The absorbance was measured at 550 nm using a Fluostar Omega microplate spectrophotometer. Viability was calculated as the percentage of living cells = (treated cell OD_550_/untreated cell OD_550_) × 100. In parallel to the MTT assay, the cells after 24 h and 48 h were counted manually to estimate the proliferation rate.

### 3.9. Migration and Invasion Assays

Both assays were performed on polyethylene terephthalate membrane cell culture inserts with a pore size of 8 μm (Sarstedt, Nümbrecht, Germany). Before seeding the cells, Matrigel (100 μg/mL, Sigma-Aldrich) was added for the invasion assays. Briefly, in the upper compartment, 10^5^ MDA, sh-MDA, or 15.10^5^ MCF-7 cells were seeded in 100 µL FBS-free Opti-MEM medium. After 24 h, each of the four sugars were added at 25, 50, or 100 µg/mL, or not and treated with a vehicle (control) in the upper compartment and the bottom compartment was filled with fresh medium supplemented with 10% FBS for 24 h at 37 °C. OGT 2115 was used at 5 μM and DMSO-treated cells as control. After incubation, the inserts were washed with PBS and cells fixed with cold absolute ethanol at −20 °C for 10 min. The remaining cells were removed from the upper side of the membrane by scrubbing with cotton swab and cells from the bottom side were stained with 0.1% crystal violet for 20 min, then captured with a Zeiss microscope 40× objective. The membranes were seeded, and crystal violet staining was eluted in 10% acetic acid and the absorbance was measured at 600 nm using a Fluostar Omega microplate spectrophotometer.

### 3.10. Chemotaxis Assay

Chemotaxis assay was performed on inserts with a pore size of 8 μm, as described above. Briefly, in the upper compartment, 10^5^ MDA, sh-MDA cells were seeded in 100 µL FBS-free Opti-MEM medium. After 24 h, Opti-MEM supplemented with 10% FBS with or without different sugars at 100 µg/mL for 8 h was added in the bottom compartment while in the upper one, 100 µL of FBS-free Opti-MEM medium was added. Inserts were then processed as described above.

### 3.11. Statistics

All the values are expressed as the mean ± SEM of the indicated number of independent cultures, as specified in the figure legends. A one-way ANOVA followed by a Fisher LSD post hoc test was used to compare more than two groups. A Student’s *t*-test was used to compare two groups. A *p*-value < 0.05 was considered significant. Statistical analysis was performed using GraphPad Prism version 5.0 (San Diego, CA, USA).

## 4. Conclusions

λ-CO is a sulphated oligosaccharide that we generated from marine red algae’s λ-CAR polysaccharide by radical depolymerisation, as previously described [24]. In this study, we deepened its mechanisms of action. We demonstrated that depolymerisation of the native λ-CAR completely reversed its unfavourable properties towards TNBC cells motility. Indeed, unlike λ-CAR, λ-CO efficiently inhibited MDA-MB-231 cell migration and invasion in a HPSE-dependent manner. These results were stronger and different from those observed with heparins, though they are better inhibitors of HPSE, advocating for the idea that the anti-invasiveness properties of λ-CO are not fully dependent on a direct inhibition of the enzymatic activity. Indeed, we found that TNBC treatment with λ-CO also reduced HPSE level and trafficking. λ-CO also downregulated additional factors linked with ECM remodelling and breast cancer progression, in particular MT1-MMP and its direct target MMP-2, and our data suggest that HPSE protein level acts as an orchestrator of MMP-2 activation. Our original work showed that HS mimetics developed to target HPSE in cancers should not be only screened relying on their in vitro anti-HPSE activity, but that cell-based experiments can reveal unexpected additional effects. We demonstrated that this original λ-CO oligosaccharide shows no apparent toxicity on TNBC cells. Interestingly, we found that TNBC treatment with λ-CO drastically reduces HPSE levels along with MMP-2 activity, a putative mechanism underlying the observed reduction of TNBC cell migration. Nevertheless, the exact mechanisms involved in the action of λ-CO on HPSE transcription remain to be explored. To conclude, our study validated a marine λ−oligocarrageenan as a valuable non-toxic molecule to target invasive phenotype of TNBC and should pave the way for the exploration of this OS as a potent migrastatic agent for new therapeutic opportunities in cancer.

## Figures and Tables

**Figure 1 marinedrugs-19-00546-f001:**
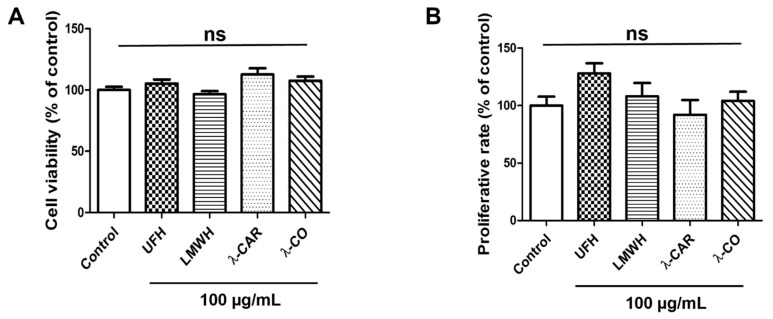
Effects of the four sugars, Unfractionated Heparin (UFH), Low Molecular Weight Heparin (LMWH), native λ-Carrageenan (λ-CAR), and depolymerized λ-CAR (λ-CO) on parental MDA-MB-231 cell viability and proliferation rate. (**A**) MTT assay was used to analyse cell viability of cells treated with vehicle (control) or compounds at 100 µg/mL for 24 h. (**B**) Cells were counted after 24 h treatment with either vehicle (control) or compounds at 100 µg/mL. Values are the mean ± SEM of three independent experiments. ANOVA followed by post hoc Fisher’s LSD test. ns: non-significant. Control: parental MDA-MB-231 cell line.

**Figure 2 marinedrugs-19-00546-f002:**
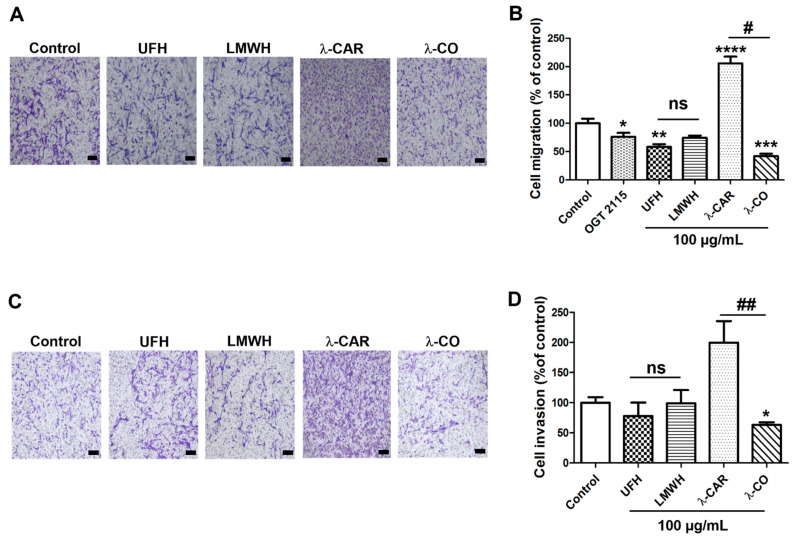
Inhibitory effects of the four sugars on MDA cell migration and invasion. (**A**) Representative microphotographs and (**B**) Quantification of MDA cell migration upon vehicle (control), compounds (100 µg/mL) or OGT 2115 (5 µM) treatment for 24 h. Scale bar in A: 100 μm. (**C**) Representative microphotographs and (**D**) quantification of MDA cell invasion upon vehicle (control) or compounds (100 µg/mL) treatment for 24 h. Scale bar in C: 100 μm. Values are the mean ± SEM of four (**A**,**B**) or three (**D**) independent experiments. * *p* < 0.05; ** *p* < 0.01; *** *p* < 0.005; **** *p* < 0.001; ANOVA followed by post hoc Fisher’s LSD (for **B**,**D**). # *p* < 0.05; ## *p* < 0.01; ns: non-significant for (**B**,**D**); ANOVA followed by post hoc Fisher’s LSD between λ-CAR and λ-CO.

**Figure 3 marinedrugs-19-00546-f003:**
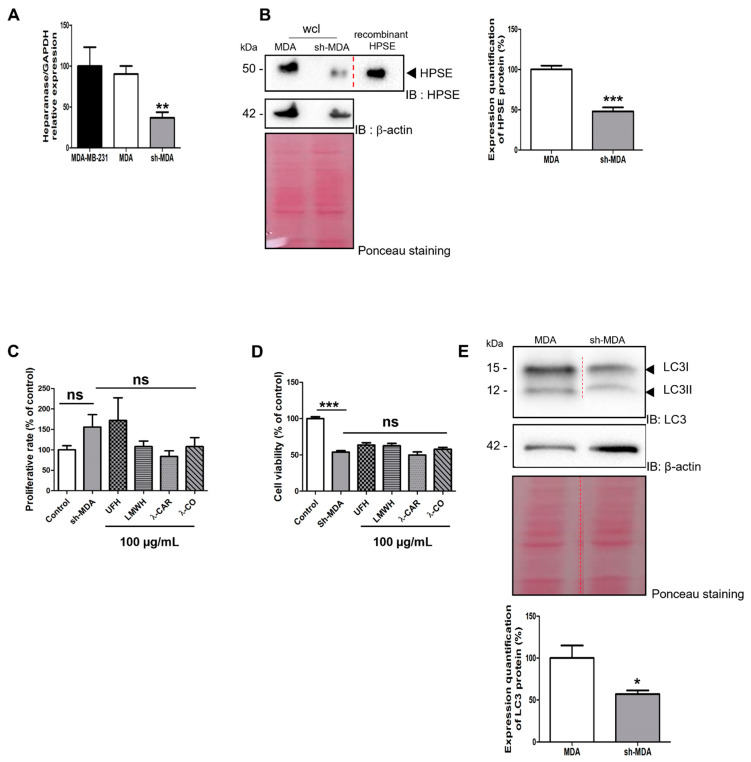
Effects of the four sugars on sh-MDA cells. (**A**) Analysis by RT-qPCR of *HPSE* gene expression 48 h after transfection of MDA-MB-231 (parental cells); MDA: transfected with a scramble shRNA; sh-MDA: transfected with sh-RNA against HPSE. (**B**) WB analysis of HPSE levels on MDA and sh-MDA cells and the corresponding quantifications normalised with β-actin. wcl: whole cell lysates. (**C**) Effects of the compounds on proliferation rate. sh-MDA cells were counted after treatment with either vehicle (control) or compounds at 100 µg/mL for 24 h. (**D**) MTT assay was used to analyse cell viability of sh-MDA cells treated with vehicle (control) or treated with compounds at 100 µg/mL for 24 h. (**E**) WB analysis of LC3 levels in MDA compared with sh-MDA and the corresponding quantifications normalised with β-actin. Ponceau S staining is shown as loading control. Values are the mean ± SEM of three independent experiments. *: *p* < 0.05; **: *p* < 0.01; ***: *p* < 0.005; ns: non-significant. ANOVA followed by post hoc Fisher’s LSD for (**A**,**C**,**E**); Student *t* test for (**B**). IB: immunoblot.

**Figure 4 marinedrugs-19-00546-f004:**
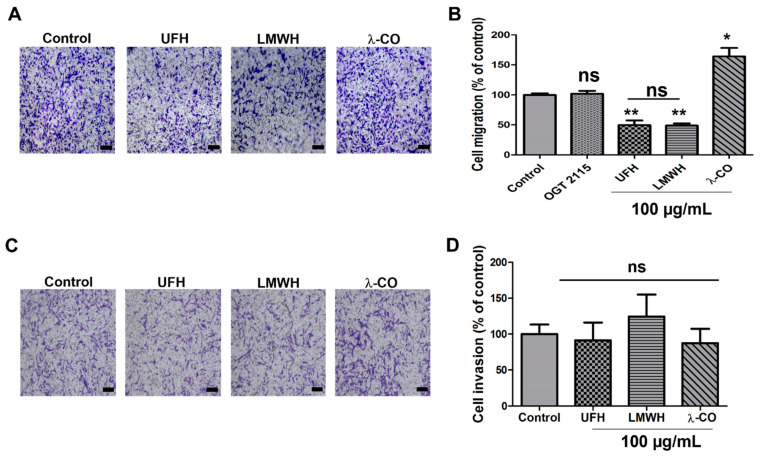
Inhibitory effects of the four sugars on sh-MDA cell migration and invasion. (**A**) Representative microphotographs and (**B**) Quantification of sh-MDA cell migration upon vehicle (control), compounds (100 μg/mL) or OGT 2115 (5 µM) treatment for 24 h. Scale bar in A: 100 μm. (**C**) Representative microphotographs and (**D**) quantification of sh-MDA cell invasion upon vehicle (control) or compounds (100 μg/mL) treatment for 24 h. Scale bar in (**D**): 100 μm. Values are the mean ± SEM of four (**A**,**B**) or three (**D**) independent experiments. * *p* < 0.05; ** *p* < 0.01; ns: non-significant; for (**B**,**D**); ANOVA followed by post hoc Fisher’s LSD (for **B**,**D**).

**Figure 5 marinedrugs-19-00546-f005:**
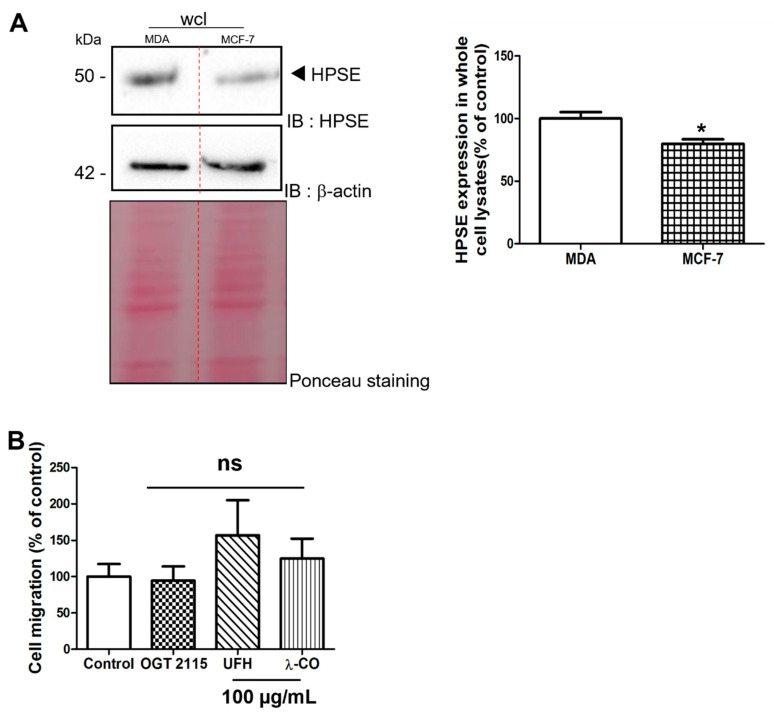
(**A**) Analysis by Western blot of HPSE levels in MDA and MCF-7 cells and the corresponding quantifications normalised with β-actin. Ponceau S staining is showed as loading control. (**B**) Quantification of MCF-7 migration upon vehicle (control), compounds (100 µg/mL), or OGT 2115 (5 µM) treatment for 24 h. Data are shown as the mean ± SEM of three independent experiments. Values are the mean ± SEM of three independent experiments. ns: non-significant, for (**B**), ANOVA followed by post hoc Fisher’s LSD. * *p* < 0.05; ns: non-significant Student *t* test for (**A**). IB: immunoblot.

**Figure 6 marinedrugs-19-00546-f006:**
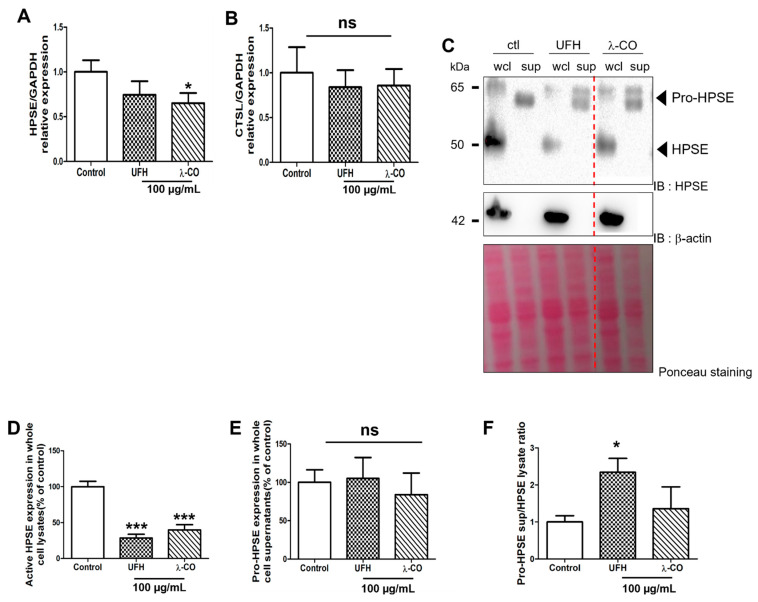
Analyses of the expression of HPSE (**A**) and cathepsin L (**B**) by RT-qPCR after 24 h treatment of MDA cells with UFH and λ-CO at 100 µg/mL. (**C**) Western blot analyses of HPSE after 24 h treatment of UFH and λ-CO at 100 µg/mL and the corresponding quantifications normalised with β-actin of active HPSE (**D**) and pro-HPSE (**E**) in whole cell lysates (wcl) and cell supernatants (sup) after 24 h treatment of UFH and λ-CO at 100 µg/mL. The red dotted line indicates splicing of non-adjacent lanes in the same WB. Ponceau S staining is showed as the loading control. (**F**) Ratio of pro-HPSE sup/HPSE wcl. Values are the mean ± SEM of three independent experiments. * *p* < 0.05, *** *p* < 0.005; ns: non-significant, ANOVA followed by post hoc Fisher’s LSD. Control: MDA cells untreated. IB: immunoblot.

**Figure 7 marinedrugs-19-00546-f007:**
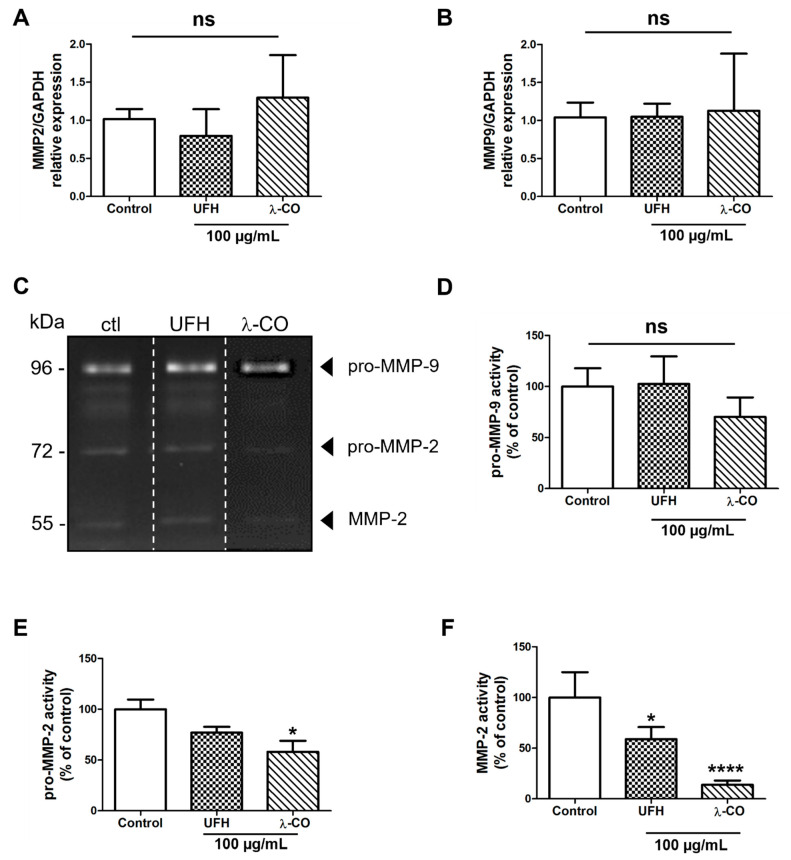
Analyses of the expression of MMP2 (**A**) and MMP9 (**B**) by RT-qPCR after 24 h treatment of MDA cells by UFH and λ-CO at 100 µg/mL. (**C**) Gelatin gel zymography images. Analyses of the activity of pro-MMP-9 (**D**), pro-MMP-2 (**E**), and MMP-2 (**F**) after 24 h treatment of UFH and λ-CO at 100 µg/mL. The white dotted line in C indicates splicing of non-adjacent lanes in the same zymogel. Data are shown as the mean ± SEM of three independent experiments. Values are the mean ± SEM of three independent experiments. * *p* < 0.05; **** *p* < 0.001; ns: non-significant, ANOVA followed by post hoc Fisher’s LSD. Ctl: Control: MDA cells untreated. ns: non-significant.

**Figure 8 marinedrugs-19-00546-f008:**
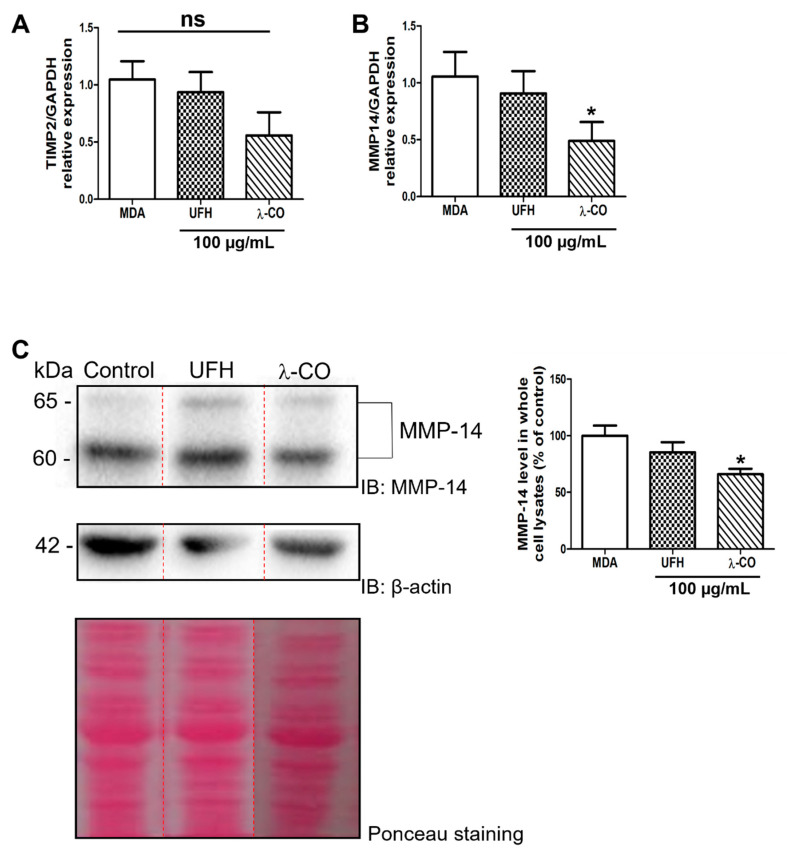
Analysis of the expression of TIMP2 (**A**) and MMP14 (**B**) encoding genes by RT-qPCR after treatment of MDA cells by UFH and λ-CO at 100 µg/mL for 24 h. (**C**) Analysis of MMP-14 levels in whole cell lysates after treatment of UFH and λ-CO at 100 µg/mL and the corresponding quantifications normalised with β-actin. The red dotted line indicates splicing of non-adjacent lanes in the same WB. Ponceau S staining is showed as the loading control. Data are shown as the mean ± SEM of three independent experiments. Values are the mean ± SEM of three independent experiments. * *p* < 0.05; ns: non-significant ANOVA followed by post hoc Fisher’s LSD. Ctl: Control: MDA cells untreated. IB: immunoblot.

**Figure 9 marinedrugs-19-00546-f009:**
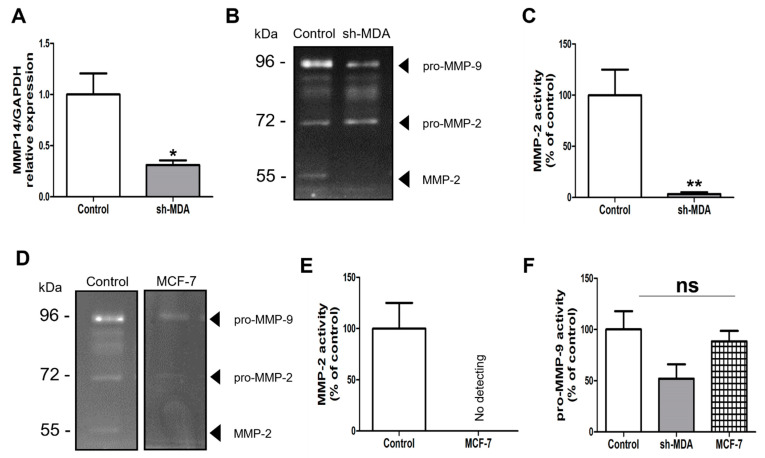
(**A**) Analysis of the expression of MMP-14 encoding gene by RT-qPCR on sh-MDA cells. (**B**) Gelatin gel zymography images and (**C**) Quantification of MMP-2 activity on MDA cells (control) compared to sh-MDA cells. (**D**) Gelatin gel zymography images showing level of MMP-2 in MDA cells compared to MCF-7 cells. (**E**) Quantification of MMP-2 activity on MDA cells (control) compared to MCF-7 cells. (**F**) Quantification of pro-MMP-9 levels in MDA cells (control) compared to sh-MDA and MCF-7 cells. Data are shown as the mean ± SEM of three independent experiments. Values are the mean ± SEM of three independent experiments. *: *p* < 0.05; **: *p* < 0.01. Student *t* test for (**A**,**C**). Control: MDA cells untreated. ns: non-significant. ANOVA followed by post hoc Fisher’s LSD for (**F**).

**Table 1 marinedrugs-19-00546-t001:** Physicochemical properties and IC50 values towards HPSE activity of Unfractionated Heparin (UFH), Low Molecular Weight Heparin (LMWH), native λ-Carrageenan (λ-CAR), and depolymerized λ-CAR (λ-CO).

Molecules	Molecular Weight (kDa)	Polydispersity Index (PI)	Sulfation Degree (%)	IC50 ^4^ (µg/mL)
UFH	12.7 ^1^	1.1	43.9 ± 1.9	0.63 ± 0.04
LMWH	5.1 ^2^	1.9	28	1.02 ± 0.08
λ-CAR	2585.9 ^1^	1.1	~30	0.12 ± 0.03 *
λ-CO	5.9 ^2^–1.2 ^3^	1.4	17.8 ± 0.9	6.0 ± 1.2

* High viscosity of native λ-carrageenan may induce unspecific inhibition. ^1^ According to the manufacturer. ^2^ MW estimated by SEC-HPLC with a calibration curve made of pullulan standards. ^3^ MW estimated by SEC-HPLC with a calibration curve made of heparin standards. ^4^ IC50 measured on recombinant HPSE [24].

**Table 2 marinedrugs-19-00546-t002:** Primers used for the qPCR analysis.

Genes	Primers Forward (5′-3′)	Primers Reverse (5′-3′)
h-HPSE	GGTCCTGATGTTGGTCAGCC	GTCCATTCAAATAGTAGTGATGCCA
h-GAPDH	GGCTCTCCAGAACATCATCCCTGC	GGGTGTCGCTGTTGAAGTCAGAGG
h-CTSL	AAGAACAGCTGGGGTGAAGAAT	CATCCCCAGTCAAGTCCTTCC
h-MMP14	ATGTGGTGTTCCAGACAAGTTTGGGG	CAAGGCTCGGCAGAGTCAAAGTGGG
h-MMP2	GCTGGGAGCATGGCGATGGATACC	GGACAGAAGCCGTACTTGCCATCC
h-MMP9	GACGCCGCTCACCTTCACTC	TTGGAACCACGACGCCCTTG
h-TIMP2	ATCAGGGCCAAAGCGGTCAGTGAG	ATCTTGCACTCGCAGCCCATCTGG

## Data Availability

The data presented in this study are available on request from the corresponding author.

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
