# Peer review of "A Marine λ-Oligocarrageenan Inhibits Migratory and Invasive Ability of MDA-MB-231 Human Breast Cancer Cells through Actions on Heparanase Metabolism and MMP-14/MMP-2 Axis"

_marinedrugs, 2021, doi:10.3390/md19100546_

Round 1
Reviewer 1 Report
Overall, this is an interesting study that builds off of the investigators’ previous work to further explore a novel oligosaccharide that affects heparanase. The study is well designed and thorough and most data support the author’s conclusions. However, there are several changes that are needed before the manuscript should be accepted for publication.
The manuscript requires editing for grammatical issues. A couple examples include:
Line 30 – It is not clear what is meant by “and that” following “compared with heparins”.
Line 30 – “We evidenced...” should be replaced with “we provided evidence...”.
Line 40 – “TNBC status is associated to...” should be replaced with “TNBC status is associated with...”.
Line 44 – “...remain the most popular cares...” should be replaced with “...remain the most popular treatments...”.
Line 184 “...sc-MCF-7 cells line” should be replaced with “...sc-MCF-7 cell line”.
Line 186 “...reach bloodstream...” should be replaced with “...reach the bloodstream...”.
The introduction should be broken up into different paragraphs.
Line 28 “using shRNA strategy” should be deleted. “sh-MDA-MB-231” given on line 27 is sufficient to let the readers know sh-RNA was the strategy used.
The data in Figure 1C should be added to Figure 1B.
Line 258. The authors could replace “seems to” with “may” in this sentence since at this point they have not yet provided evidence that the mitochondria are involved.
For Figure 3 the authors need to include graphs (expression quantification of b-actin protein (%)) for b-actin levels in parts B and E. The levels of b-actin shown in the western blots suggests that protein loading was not the same for the different samples so graphs for b-actin are needed to help put the HPSE and LC3 graphs into context.
The data in Figure 4C should be added to Figure 4B.
The data in Figure 5C should be added to Figure 5B.
For Figure 5 the authors need to include a graph (b-actin expression in whole cell lysates (% of control)) for b-actin levels in part A. The levels of b-actin shown in the western blot suggests that protein loading was not the same for the different samples so a graph for b-actin is needed to help put the HPSE graph into context.
Lines 341 and 342. The authors should spell out whole cell lysates and cell supernatants.
Lines 351-355. The authors are perhaps making too strong of a conclusion given the data. The authors state that the effects of l-CO seems more related to a down-regulation of HPSE gene expression and UFH is affecting trafficking, but the data do not seem so clear to this reviewer. For instance, both UFH an l-CO induced a decrease in gene expression and trafficking; albeit at slightly different levels.
Lines 405-406. The authors are making too strong of a conclusion given the data. No experiments were conducted to show MMP-14 directly affected MMP-2 activation.
For Figure 8 the authors need to include a graph (b-actin level in cell lysates (% of control)) for b-actin levels in part C. The levels of b-actin shown in the western blot suggests that protein loading was not the same for the different samples so a graph for b-actin is needed to help put the MMP-14 graph into context.
Lines 433-434. The authors mention a previous study confirmed that in MDA-MB-231 HPSE overexpression decreased MMP-14 and MMP-2. These data appear to be the opposite of what the authors found with the same cell line. This should be discussed further.
Reviewer 2 Report
The manuscript describes the fact of marine oligocarrageenan as a potent migrastatic agent for new therapeutic opportunities in cancer. The findings are very interesting, experimental strategy is very elegant. The paper is written in a good, accessible language.
I recommend the acceptance of the manuscript after minor considerations.
- Legend to figure 1, Line 172: “ns: non-significant” written twice.
- All western blots should be done again as actin bands are differ too much. In this case, it is not correct to compare data.
Author Response
We warmly thank Reviewer 2 for its positive feedbacks and support about the study. The minor modification about legend in Figure 1 has been made accordingly. We fully understand the Reviewer 2 concerns about the western blot that have also been raised by another reviewer. We are aware of the possibility of misinterpretation with this technique and understand that visual of the beta-actin level may be confusing when further discussing expression of protein of interest in different conditions. To lift any confusion, we have therefore included in the revised version ponceau S staining as loading control under all the representative western blot of the manuscript but also all the beta-actin quantification in a supplementary figure (figure S2).
Reviewer 3 Report
The present study aims to investigate the effect of a derivative of a marine oligosaccharide, named λ-CO, in TNBC cells. Since sugar-based molecules like heparins or sulfated polysaccharides of similar structure have been developed and studied for controlling HPSE activity, the authors used established TNBC cell lines and cell lines where HPSE was knocked down. They have examined cell viability, proliferation, migration, and invasion. Moreover, they have examined the effect of λ-CO on gelatinases and MMP-14 expression. The results obtained suggest that λ-CO was more efficient than heparins to reduce cell migration and invasion, but in a HPSE dependent-manner. Moreover, they also suggest that λ-CO tightly controlled a HPSE/MMP-14/MMP-2 axis leading to reduced MMP-2 activity. Therefore, the authors concluded that λ-CO seems to possess anti-cancer activities through its dual effect on HPSE, i.e., by reducing its enzymatic activity and by affecting the functions controlled by the HPSE levels.
The study is interesting, however, there are many points requiring correction and/or clarification before considering the manuscript for publication.
Major points
- Extensive linguistic revision is required (i.e., tumor vs tumour). Use past tense in results. Care should be taken in the names of cell lines. To help the reader use similar symbols for cells subjected to similar treatment (sc-MCF-7 vs MDA).
- Extensive revision in figures 3, 5, 6, and 8 is required. Panels showing ponceau S staining should be moved to the supplementary and in almost all panels the graphs should be revised to have in axes’ legends letters of similar and larger size. Also decrease the free space. Moreover, use the correct names of cells the axes’ legends (see, i.e., figure 5).
- It is very important to examine λ-CO for any effect on HPSE expression.
- Lines 305-307: The authors should clearly describe between what cells they did the various comparisons. Why they did not examine any differences between sc-MCF-7 and MCF-7 cells? The difference in HPSE expression was less than 20%, thus it is difficult to obtain clear conclusions.
Minor points
- It is preferable to use the term compounds (or derivatives) and not candidates.
- Use subheadings for heading 2.2 and organize the text appropriately.
- Lines 187-188: Delete this sentence.
- Lines 253-255: This statement should be deleted or revised appropriately.
- Lines 424-429: Explain and divide to two sentences.
- Lines 434-436: Revise carefully; there are missing words.
- Line 439: Use a more appropriate word instead of “cell model”.
- It is suggested to revise all paragraph between lines 419 & 445. It is not clear whether the authors are referred to cells or tissues. It is not clear what studies are compared.
Reviewer 4 Report
Cousin et al. study the mechanism by which l-oligocarrageenan decreases the invasive ability of breast cancer cells by using the triple negative MDA-MB-231 and the estrogen-receptor positive MCF-7 cell line. To investigate the role of heparinase, silencing of the gene was performed and transfected, scramble RNA transfected, not transfected MDA-MB-231 and MCF-7 cells studied.
Specific major comments
- Different cells were used in the study and viability was affected by the transfection. Therefore, it would be good to show cytotoxicity of the compound not only in the non-transfected MDA-MB-231 cells but also in the other cells.
- The effect of autophagy on tumor progression is stage-dependent. Maybe this should be added.
- The authors used a concentration of 100 µg/ml in the study because it inhibited migration in MDA-MB-231according to a previous publication. In another publication, the authors showed inhibition of angiogenesis at 200 µg/ml only. Is there any increase in the anti-migratory effect at 200 µg/ml?
Round 2
Reviewer 3 Report
The authors have followed most of the comments raised and the manuscript can be accepted for publication.
Reviewer 4 Report
The authors addressed my comments.